# Association between Frailty and Asthma

**DOI:** 10.3390/medicina60091479

**Published:** 2024-09-10

**Authors:** Jong Myung Park, Sujin Lee, Jae Ho Chung

**Affiliations:** 1Department of Thoracic and Cardiovascular Surgery, Pusan National University School of Medicine, Busan 49241, Republic of Korea; balmuda@gmail.com; 2Department of Thoracic and Cardiovascular Surgery, Pusan National University Yangsan Hospital, Yangsan 50612, Republic of Korea; 3Transplantation Research Center, Research Institute for Convergence of Biomedical Science and Technology, Pusan National University Yangsan Hospital, Yangsan 50612, Republic of Korea; 4Department of Neurology, International St. Mary’s Hospital, Catholic Kwandong University College of Medicine, Incheon 22711, Republic of Korea; neuroneuro@naver.com; 5Department of Internal Medicine, International St. Mary’s Hospital, Catholic Kwandong University College of Medicine, Incheon 22711, Republic of Korea

**Keywords:** asthma, elderly, frailty

## Abstract

*Background and Objectives*: This study investigated whether there is an association between elderly frailty and asthma. *Material Methods*: We examined 9745 elderly participants who did not have asthma and 275 elderly patients who had asthma diagnosed by a doctor from the 2020 Survey of Living Conditions and Welfare Needs of Korean Older Persons Survey. Study Selections: The Korean version of the fatigue, resistance, ambulation, illnesses, and loss of weight (K-FRAIL) scale was used to determine their level of frailty. The relationship between frailty and geriatric asthma was examined using multiple logistic regression analysis, which was adjusted for a number of confounding variables (socioeconomic, health behavior, psychological characteristics, and functional status). *Results*: Frailty as defined by the K-FRAIL scale was significantly higher in the asthma group (7.6%) than the non-asthma group (4.9%). The frailty phenotype component showed that resistance, ambulation, and illness severity were more severe in the asthma group than the non-asthma group. After adjusting, asthma was significantly associated with an increased risk of frailty (OR 1.45; 95% confidence interval [CI] 1.01–2.09) compared to the non-asthma group. *Conclusions*: Frailty might be associated with elderly asthma in patients from the Korean population. Frailty may not only be associated with asthma, but also with other diseases. So, more evidence is needed to establish this association.

## 1. Introduction

Frailty is the deterioration of the homeostatic reserve of the body’s organs with the aging process [1]. As the physiological reserve function decreases due to aging, so too does the body’s performance and cognitive function, while co-morbidities increase. Eventually frailty develops, leading to functional dependence, the worsening of disease, increased hospitalization rates, and increased mortality [2]. Many variable frailty criteria and indices have been evaluated, so an appropriate methodology that is easy-to-use in clinical practice to detect frailty is crucial in geriatric research. The fatigue, resistance, ambulation, illnesses, and loss of weight (FRAIL) scale is a screening tool for frailty status using a simple five-item questionnaire [3] and a Korean version of the FRAIL scale (K-FRAIL) has been well validated in Korean populations [4].

Although our study did not include COPD patients, Marengoni et al. showed that COPD patients had a twofold increased risk of frailty in a meta-analysis [5]. Moreover, elderly COPD with frailty had poorer survival and a poorer quality of life than elderly COPD without frailty [6]. Regarding asthma, there are few studies which investigate the association of frailty in asthma patients [7,8,9].

We postulated a hypothesis that there will be a high association between frailty and asthma since both illnesses share risk factors (tobacco use and aging) and mechanisms (inflammatory cytokines and endocrine dysfunction). To our knowledge, there have been no studies on the association between asthma and frailty assessed using the K-FRAIL scale in the Korean population. The cited studies differed epidemiologically, and therefore ethnicity-specific evaluations are necessary. Understanding the frailty risk of people with elderly asthma is vital for the accurate evaluation of health interventions for health providers. Thus, utilizing data from a nationally representative Korean elderly population, we examined whether frailty as measured by the K-FRAIL scale and elderly asthma are related.

## 2. Materials and Methods

### 2.1. Study Participants

The 2020 Korea Institute for Health and Social Affairs Survey of Living Conditions and Welfare Needs of Korean Elderly People was carried out by the Korea Institute for Health and Social Affairs and the Korean Ministry of Health and Welfare. The survey’s sample size is representative of South Korean elderly. The information in the data covers a wide range of life areas, including social life, family structure, financial status, and mental and physical health. A two-phase stratified cluster selection technique was employed to choose a sample of roughly 10,000 senior citizens from 17 South Korean provinces. Each survey participant was given a sampling weight to ensure that the results were applicable to the elderly population. There was no need for further ethical approval because all participants provided informed consent and our data were made public [10]. Among 10,097 participants who were ≥65 years old in age, 324 elderly people who had not finished the Korean version of the geriatric depression scale (K-SGDS), the Korean version of the Mini-Mental Status Examination (MMSE-KC), the limited activities of daily living (ADL) scale, or the limited instrumental activities of daily living (IADL) scale were not included in our analysis. After exclusion, 9920 people (males, *n* = 3971; females, *n* = 5949; age: 65–99 years-old) made up the final analysis. Individuals who answered “Have you been diagnosed with asthma by a doctor?” affirmatively were considered patients with asthma.

### 2.2. Assessment

#### Features Linked to Health and Sociodemographics

This study examined gender (male or female), age, residence (urban or rural), living status (live alone or live with family), education duration (0–3 years, 4–6 years, or ≥7 years), job status (employed or unemployed), economic status, family income (quartile of annual household income; 1 = the lowest 25% and 4 = the highest 25%), and self-rated health (poor, moderate, good); each clinical scale used in our study had been validated in the Korean population in previous study [11]. Physical activity was defined using the International Physical Activity Questionnaire—Short Form [12], which had previously been validated in the Korean elderly community. Physical activity was defined as more than 150 min per week of exercise [13]. The definition of alcohol usage was “drinking alcohol ≥ two days/week”. The number of chronic diseases was determined based on diagnoses provided by a doctor and included diseases that had persisted for 3 months or longer and current treatments, and the total count of chronic diseases included hypertension, stroke, hyperlipidemia, ischemic heart disease, etc.; our study categorized the chronic illness variable into no chronic illness or 1 and more than 2. Nutrition was assessed using the Nutrition Screening Initiative’s “Determine Your Nutritional Health” questionnaire [14]. The overall score of 10 items is classed as 0–2 (good nutrition), 3–5 (moderate nutritional risk), and ≥6 (high nutritional risk). Fall injury was defined if they reported any falls in the past 12 months before the survey.

### 2.3. Elderly Depression (Korean Version of the 15-Item Geriatric Depression Scale)

The geriatric depression scale (GDS) was created to evaluate elderly experiences of depression, decreased activity, irritability, withdrawal and pain, and negative perspectives of the past, present, and future [14]. It is a widely used tool for screening for depression in older people who live in a community such as our national representative population. The GDS is a short form composed of 15 questions that only takes around four minutes to complete; when compared to the results of structured clinical interviews for the diagnosis of depression, it has a sensitivity of 85–92% and a specificity of 65–81% [14]. The Korean version of the GDS (K-SGDS) was also a reliable method [15]. Since 8 is the suggested ideal cut-off value for the K-SGDS [16], respondents were categorized into two groups: those who were depressed (≥8) and those who were not (<8).

### 2.4. Functional Capacity

This study examined activities of daily living (ADL) and Instrumental Activities of Daily Living (IADL), as well as hearing, visual, and chewing impairment. Using the Korean ADL scale [17], ADL were assessed. Individuals were classed as having an ADL limitation if they had limits in more than one of the following categories: 1. putting on clothes; 2. cleaning one’s face, teeth, and hair; 3. taking a bath; 4. eating; 5. getting up from a sitting position and moving around the room; 6. using the toilet; and 7. controlling one’s bowels and bladder.

The Korean Instrumental ADL Scale was used to define IADL limits [17]. IADL was defined as the incapacity to complete any ten of the following tasks: 1. personal hygiene; 2. household chores; 3. meal planning; 4. laundry; 5. punctual medication intake; 6. financial management; 7. short-distance travel; 8. making decisions about what to buy, how much to pay, and how to accept change; 9. communicating by phone; and 10. utilizing public transportation.

### 2.5. Evaluation of Frailty

The FRAIL scale is a 5-item questionnaire (fatigue, resistance, ambulation, illness, and loss of weight) [15] which has suggested easy screening indices for assessing frailty [3] and has no requisite for physical examination; a Korean version of the FRAIL scale (K-FRAIL) has also been validated in the Korean population [4].

Fatigue was defined using answers to the following question: “Have you recently been much less active or felt less motivated to be active?” Resistance was defined by asking the following question: “Do you have difficulty climbing 10 steps without a rest?” Ambulation was defined using the following question: “Do you have difficulty walking 400 m without an assistive device?” Illness was defined by asking the following question: “Have you been diagnosed with a chronic disease by your physician?” (0–4 comorbidities = 0, ≥5 comorbidities = 1). Finally, weight loss was determined by asking the following question: “Have you experienced weight loss of more than 5% in the last year?” Based on the total scores, the participants were defined as non-frail (0 point), pre-frail (1–2 points), or frail (3–5 points). The K-FRAIL scale is typically used to assess frailty risk rather than to diagnose frailty. It is a validated tool to identify individuals who are at risk of frailty by evaluating specific criteria, but it is not a diagnostic tool for frailty.

### 2.6. Data Analysis

Descriptive statistical methods were used to describe the research population’s basic characteristics, with numbers and percentages provided for each variable. Living and household conditions, educational level, family income, and depression affect frailty [18]; we therefore made adjustments for health behavioral variables (such as smoking, drinking, having a good diet, and engaging in regular exercise), sociodemographic variables (such as age, sex, residence area, living status, employment, BMI, education level, economic position, fall injury, hospitalization in previous year, and co-morbidities), psychological variables (i.e., health status, depression, and life satisfaction), and functional status (cognitive impairment, visual, hearing, or chewing impairment, ADL, IADL, and leg muscle weakness). All statistical analyses were created using the R programming language (R version 4.3.2) and a variety of tools. *p* < 0.05 was regarded as statistically significant.

## 3. Results

### 3.1. Baseline Characteristics of Study Populations

Table 1 shows the sociodemographic characteristics of the participants. The asthma group showed significantly higher rates of having no spouse, living alone, being unemployed, being less educated, smoking more, suffering from more chronic diseases, having bad self-rated subjective health, suffering from more nutritional risk, and using more medication drugs; depression was significantly more common in the asthma group than the non-asthma group. These findings might suggest that relatively poor socioeconomic status and bad mental health problems are more prevalent in elderly South Korean patients with asthma than in those without asthma.

### 3.2. Functional Status of Study Populations

Functional status variable differences between the asthma and non-asthma participants are shown in Table 2. The rates of visual, hearing, or chewing impairment, ADL, IADL independence, and leg muscle weakness were significantly higher in the asthma group than the non-asthma group. These findings might suggest that elderly South Korean asthma patients have relatively worse functional status. Frailty as defined by the K-FRAIL scale more was significantly higher in the asthma group (7.6%) than in the non-asthma group (4.9%) (Figure 1). The frailty phenotype component showed that resistance, ambulation, and illness severity were more severe in the asthma group than in the non-asthma group.

### 3.3. Fraility Risk According to Asthma Status

After adjusting for multiple confounding variables, asthma was significantly associated with an increased risk of frailty (OR 1.45; 95% confidence interval [CI] 1.01–2.09) compared to the non-asthma group (Table 3). These findings might suggest that asthma was associated with frailty regardless of multiple confounding variables.

## 4. Discussion

Using a large sample of community-living Korean senior people, we investigated the association between frailty status using the K-FRAIL scale in elderly asthma patients from the Korean population, and we discovered that frailty and elderly asthma in patients from the Korean population were closely associated. Frailty is a well-known independent risk factor of chronic respiratory diseases such as COPD, and chronic respiratory diseases can lead to frailty [17]. Frailty has a chronic respiratory disease incidence ranging from 5% to 65%. Physical inactivity due to breathlessness in chronic respiratory disease patients increases the prevalence of frailty [19]. In our study, the prevalence of frailty was 14.6% of the participants (mean age: 73.4 years), which is almost compatible with another study [19], although direct comparison is difficult because of differences in methodologic frailty criteria variability. Frailty is also related to poor outcomes in chronic respiratory disorders, including more falls, hospitalizations, and higher levels of impairment [18,20] and risk of mortality [21].

Our study showed asthma was associated with frailty, which is in line with findings from other studies [7,8,9]. Ma et al. suggested a probable causal effect of asthma on the risk of developing frailty, potentially mediated by reduced physical activity endurance [7]. Our study showed that frailty prevalence was 7.6%, which is similar to Verrduri’s study (9.5%) [22]. A simple tool of hand grip strength measurement for frailty resulted in an AUC of 71.6% (61.5–80.4%; *p* < 0.002), as well as a sensitivity of 73.58% and a specificity of 67.53% [23]. Another cross-sectional observation study showed that 15.4% of elderly asthma (>65 years old) patients were considered as being frail [8]. Community-dwelling adults of the GAZEL cohort showed that asthma patients had an increased risk of frailty compared with those without asthma [9], which is compatible with our study’s findings.

The mechanisms between frailty and asthma have not been fully elucidated, but the following mechanisms are suggested. Inflammation is one of the possible pathways in the association between elderly asthma and frailty [24]. Dysregulated inflammatory cytokines such as IL-6 are involved with variable patterns of the aging process and associated with decreased muscle strength, mobility, falls, frailty, and mortality [25]. There are physiobiological changes such as hypoxia, steroid use, and malnutrition which contribute to frailty. Hypoxia can generate systemic inflammation which causes decreased muscle strength, muscle endurance, and muscle mass (sarcopenia), as well as increased muscle weakness [26,27]. Decreased physical activity is associated with exacerbations and hospitalizations [28], all of which add to disuse muscle atrophy and raise the likelihood of developing skeletal muscle dysfunction and frailty. Steroids are often prescribed for elderly asthma patients who experience asthma exacerbations; steroid use, which causes muscle wasting, eventually results in an increase in frailty [29]. Malnutrition is also a risk factor of frailty in chronic lung diseases such as asthma [30], so we adjusted nutritional status. Food intake in patients with lung diseases such as asthma can cause dyspnea aggravation. So, chronic lung diseases such as asthma may decrease food intake to avoid this experience. An increased metabolic rate coupled with reduced nutritional intake might cause patients with chronic lung diseases such as asthma patients to be at an increased risk of malnutrition [30]. There may be a reciprocal association between frailty and chronic respiratory conditions. According to Vaz Fragoso et al., respiratory impairment was linked to increasing frailty and frailty was linked to higher respiratory impairment [21]; this bidirectional pathophysiology may have significant ramifications because treatments for one ailment may also benefit the other. Subcomponents in the K-FRAIL scale have distinct characteristics. The fatigue item is associated with depression rather than physical performance. As shown in the original study by Morley et al. [3], the illness subcomponent had a low incidence and the weight loss subcomponent was associated with a risk of malnutrition. In our study, the illness subcomponent of frailty was associated with the strongest risk of frailty in elderly asthma patients. These contradictory findings may have resulted from methodological differences. In spite of frailty’s importance for assessing health outcomes (to increase mortality, improve poor quality of life, etc.), clinicians in Korea face barriers for using frailty criteria. The K-FRAIL scale can be quickly applied by patients or by a physician. Moreover, interpreting the results of this scale (robust vs. prefrail vs. frail) is easy for most clinicians. So, the K-FRAIL scale can be a useful screening method in most busy clinical practices such as those in Korea. Finally, the K-FRAIL scale is used to assess frailty risk rather than to diagnose frailty, so it is not a diagnostic tool for frailty. Our study has some limitations. First, this study’s cross-sectional design makes it difficult to draw conclusions as to whether the association between asthma and frailty involves a cause-and-effect relationship. Second, defining asthma in our study was not based on a set of standardized questions to define asthma but based on self-reported physician-diagnosed asthma only, probably leading to declaration and misclassification bias and under-evaluation of the true prevalence of elderly asthma. In addition, we did not assess current asthma symptoms and corticosteroid usage, or the rate of asthma exacerbations and hospitalizations due to asthma. Finally, the K-FRAIL scale is used to assess frailty risk rather than to diagnose frailty, so it is not a diagnostic tool for frailty. Therefore, caution is necessary when interpreting our results regarding the association between elderly asthma and frailty. Further well-designed studies are needed to elucidate the relationship between frailty and elderly asthma.

## 5. Conclusions

In conclusion, the results of this investigation indicated a correlation between frailty and elderly asthma patients from the Korean population. Frailty may not only be associated with asthma, but also with other diseases. So, more evidence is needed to establish this association. Still, this finding might help establish policies for reducing the burden of asthma and frailty in the elderly.

## Figures and Tables

**Figure 1 medicina-60-01479-f001:**
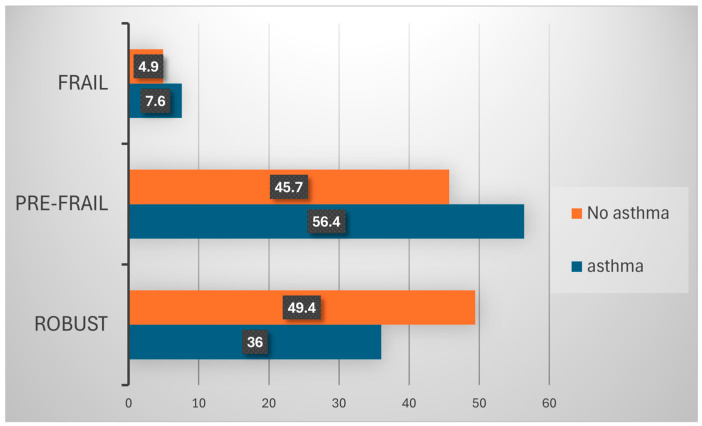
Frailty status according to asthma.

**Table 1 medicina-60-01479-t001:** Differences in sociodemographic and clinical characteristics between asthma and non-asthma patients.

	No Asthma (*n* = 9498)	Asthma (*n* = 275)	*p*-Value
Sex			<0.001
Male	5722 (60.2)	136 (49.5)	
Female	3776 (39.8)	139 (50.5)	
Age	73.4 ± 6.5	74.9 ± 6.2	0.089
Residence			0.278
Rural	2715 (28.6)	70 (25.5)	
Urban	6783 (71.4)	205 (74.5)	
Religion			1.000
Have religion	5590 (58.9)	162 (58.9)	
No religion	3908 (41.1)	113 (41.1)	
Living status			0.041
Live alone	2966 (31.2)	102 (37.1)	
Live with family	6532 (68.9)	173 (62.9)	
Job			0.014
Employed	3631 (38.2)	85 (30.9)	
Unemployed	5867 (61.8)	180 (69.1)	
BMI	23.6 ± 2.6	23.8 ± 2.7	0.089
BMI category			0.165
Underweight	285 (2.2)	6 (2.2)	
Normal	3655 (38.5)	99 (36.0)	
Overweight	3135 (33.0)	81 (29.5)	
Obese	2503 (26.4)	89 (32.4)	
Education, years			0.320
0–3	1078 (11.3)	28 (10.2)	
4–6	3116 (32.8)	102 (37.1)	
≥7	5304 (55.8)	145 (52.7)	
Economic status			0.829
1st quartile	2375 (25.0)	65 (23.6)	
2nd quartile	2380 (25.1)	69 (25.1)	
3rd quartile	2374 (25.0)	66 (24.0)	
4th quartile	2369 (24.9)	75 (27.3)	
Smoking	1030 (10.8)	42 (15.3)	0.024
Alcohol	1336 (14.1)	41 (14.9)	0.661
Regular exercise	3472 (36.6)	97 (35.3)	0.703
Chronic illness			<0.001
0–1	4502 (47.4)	32 (11.6)	
≥2	4996 (52.6)	243 (88.4)	
Subjective health status			<0.001
Good	4815 (50.7)	58 (21.1)	
Moderate	2951 (31.1)	120 (43.6)	
Bad	1732 (18.2)	97 (35.3)	
Life satisfaction			<0.001
Satisfied	8634 (93.0)	239 (86.9)	
Unsatisfied	664 (7.0)	36 (13.1)	
Cognitive impairment			0.427
Normal	6578 (69.3)	184 (66.9)	
Impairment	2920 (30.7)	91 (33.1)	
Nutrition			<0.001
Low nutritional risk	8188 (86.0)	183 (66.5)	
Moderate nutritional risk	1139 (12.0)	83 (30.2)	
High nutritional risk	191 (2.0)	9 (3.3)	
GDS (score ≥ 8)	1226 (12.9)	52 (18.9)	0.008

The short version of the GDS contains 15 questions; GDS scores range from 0 to 15, with a score ≥8 indicating depression.

**Table 2 medicina-60-01479-t002:** Differences in functional status between asthma and non-asthma patients.

	No Asthma (*n* = 9745)	Asthma (*n* = 175)	*p*-Value
Visual impairment	3108 (32.7)	131 (47.6)	<0.001
Hearing impairment	2162 (22.8)	107 (38.9)	<0.001
Chewing impairment	3539 (37.3)	160 (58.2)	<0.001
Activity of daily living (ADL)			<0.001
Independent	9114 (96.0)	248 (90.2)	
Dependent	384 (4.0)	27 (9.8)	
Instrumental ADL (IADL)			<0.001
Independent	8596 (90.5)	196 (71.3)	
Dependent	902 (9.5)	79 (28.7)	
Leg muscle weakness	2492 (26.2)	104 (37.6)	<0.001
Frailty			<0.001
Robust	4686 (49.4)	99 (36.0)	
Pre-frail	4344 (45.7)	155 (56.4)	
Frail	468 (4.9)	12 (7.6)	
Frailty phenotype component			
Fatigue	2761 (29.1)	86 (31.3)	0.420
Resistance	2170 (22.8)	97 (35.3)	<0.001
Ambulation	1930 (20.3)	77 (28.0)	0.003
Illness	14 (0.1)	9 (3.3)	<0.001
Weight loss	277 (2.9)	9 (3.3)	0.715
Fall injury	594 (6.3)	28 (10.2)	0.012
Hospitalization previous year	606 (6.4)	34 (12.4)	<0.001

**Table 3 medicina-60-01479-t003:** Adjusted Odds Ratio of asthma for frailty.

	OR (95% Confidence Interval)
Frailty	1.45 (1.01–2.09)

## Data Availability

The data that support the findings of this study are openly available from the 2020 KIHASA at https://www.kihasa.re.kr/en/publish/hsw/view?seq=38350&volume=38338 (accessed on 1 January 2020).

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
