# Peer review of "Association between Frailty and Asthma"

_medicina, 2024, doi:10.3390/medicina60091479_

Round 1

Reviewer 1 Report

Comments and Suggestions for Authors

This manuscript by Jong Myung Park et al., presents a “ Association between Frailty and Asthma” Personally,  it looks like this manuscript is not a final version.  The conclusion is not clear even the discussion is also not enough. The result part also shows just a display of each case.  

Line 30: “Conclusion: Frailty is associated with Korean elderly asthma”. – Frailty may not only be asthma but also may other diseases too. How much is related between frailty and asthma still needs more evidence.  Frailty is associated with asthma but still hard to say it is the main factor of elderly asthma. 

Line 50 missing “ ) ”

Line 66: what is “ K-SGDS”?

Line 76: what does “education level”?

Line 109 : Function capacity section: categories numbering style needs change, it is confusing with reference numbering. It is the same thing on line 134 too.

Line 121: (2009) need changed with to just the number “9”

Line 161 -162 : table 3 doesn’t need to make it. It is already mentioned in the main body.

Regarding the Result: make a more visualized data display such as a bar graph or circle graph not only a table.

The author’s contribution section showed X.X. Y.Y.  and Z.Z. What kind of things did each author do? And Institution review board section also protocol code is XXX. So authors need final check each other and make sure to submit the manuscript. This is the basic thing and it is a big mistake. 

Author Response

This manuscript by Jong Myung Park et al., presents a “ Association between Frailty and Asthma” Personally,  it looks like this manuscript is not a final version.  The conclusion is not clear even the discussion is also not enough. The result part also shows just a display of each case.  

Line 30: “Conclusion: Frailty is associated with Korean elderly asthma”. – Frailty may not only be asthma but also may other diseases too. How much is related between frailty and asthma still needs more evidence.  Frailty is associated with asthma but still hard to say it is the main factor of elderly asthma. 

Answer) I corrected as follows

Frailty is might associated with elderly asthma. Frailty may not only be asthma but also may other diseases too. So more evidences were needed to establish this association

Line 50 missing “ ) ”

Answer) I corrected

Line 66: what is “ K-SGDS”?

Answer) I corrected

Line 76: what does “education level”?

Answer) I corrected

Line 109 : Function capacity section: categories numbering style needs change, it is confusing with reference numbering. It is the same thing on line 134 too.

Answer) I corrected

Line 121: (2009) need changed with to just the number “9”

Answer) I corrected

Line 161 -162 : table 3 doesn’t need to make it. It is already mentioned in the main body.

Answer) I corrected

Regarding the Result: make a more visualized data display such as a bar graph or circle graph not only a table.

 Answer) I corrected

The author’s contribution section showed X.X. Y.Y.  and Z.Z. What kind of things did each author do? And Institution review board section also protocol code is XXX. So authors need final check each other and make sure to submit the manuscript. This is the basic thing and it is a big mistake. 

Answer) I corrected

Reviewer 2 Report

Comments and Suggestions for Authors

The topic is significant; the authors conveyed an overall well-prepared paper. The process of selection of the subjects was clear. The variables are well defined and measured appropriately. The study methods are valid and reliable. There are enough details provided in order to replicate the study.

Specific comments on weaknesses of the article and what could be improved:

Major points  

1.          The main drawback of the paper are the results section. The text in the results do not add to the data. Statistically significant results are not clear. It is not clear which results are with practical meaning. The authors should try to extend this section without repeating the tables. For example, after this, "The prevalence and adjusted odds ratios (ORs) for frailty according to asthma status 161 are presented in Table 3." the authors should describe some of the results in the text without interpreting them.

2.         It is difficult to receive information only through text and tables; the authors should present proper data through figures/graphs when applicable.   

3. The discussion is weak because the results are not shown properly. Discussion of own results lack.    

4.          The conclusion should be more strong and assertive. No valid conclusions based on the results are made. Please revise. 

Minor points

1.           Please, comment on the limitations of the study - how can they impact on your outcomes

2. Could you please discuss the clinical implications of the results and what recommendations could be made based on the results?

3.          The abstract should be one paragraph (please revise), structured (this is done)

4 The conclusion in the abstract should be revised since there is no such term as "Korean asthma" or another nationality-asthma. I suggest something like "in asthma patients from the Korean population", etc.

5. Lne 44-47 - COPD has been mentioned out of the blue; please focus on asthma OR, if you include COPD, explain in detail why you are meeting it.

6. The hypothesis in the introduction should be extended. Also, the aim should be more precise and detailed. 

7. Author contribution is not filled. 

8. State the Institutional Review Board Statement at the end of the manuscript

Author Response

The topic is significant; the authors conveyed an overall well-prepared paper. The process of selection of the subjects was clear. The variables are well defined and measured appropriately. The study methods are valid and reliable. There are enough details provided in order to replicate the study.

Specific comments on weaknesses of the article and what could be improved:

Major points  

  1. The main drawback of the paper are the results section. The text in the results do not add to the data. Statistically significant results are not clear. It is not clear which results are with practical meaning. The authors should try to extend this section without repeating the tables. For example, after this, "The prevalence and adjusted odds ratios (ORs) for frailty according to asthma status 161 are presented in Table 3." the authors should describe some of the results in the text without interpreting them.

Answer) I corrected as follows

  1. Results

Baseline characteristics of study populations

Table 1 showed sociodemographic characteristics of participants. In asthma group showed significantly higher rates of no spouse, live alone, unemployed, less educated, more smoking, more chronic diseases, bad self-rated subjective health, more nutritional risk, more medication drug and more depression were significantly more common in the asthma group than non-asthma group. These findings might suggest that relatively poor socioecomic status and had bad mental problem in elderly South Korean asthma patients than without asthma.

Functional status of study populations

Functional status variables differences between asthma and non-asthma participants were showed in Table 2. The rates of visual, hearing, chewing impairment, ADL, IADL independence, leg muscle weakness were significantly higher in the asthma group than non-asthma group. These findings might suggest that relatively bad functional status in elderly South Korean asthma patients than without asthma. We showed frail group defined by K-FRAIL scale more was significantly higher in asthma group (7.6%) than non-asthma group (4.9%) (Fig. 1). Frailty phenotype component showed that resistance, ambulation, illness severity were more severe in asthma group than non-asthma group.

Figure 1. Frailty status according to asthma

The prevalence and adjusted odds ratios (ORs) for frailty according to asthma status are presented in Table 3.

Fraility risk according to asthma status

After adjusting for multiple confounding variables, asthma was significantly associated with an increased risk of frailty (OR 1.45; 95% confidence interval [CI] 1.01–2.09) compared to non-asthma group (Table 3). These findings might suggest that asthma was associated with frailty regardless of multiple confounding variables.

  1. It is difficult to receive information only through text and tables; the authors should present proper data through figures/graphs when applicable.   

Answer) I corrected

  1. The discussion is weak because the results are not shown properly. Discussion of own results lack.    

Answer) I corrected

  1.  

  1.         The conclusion should be more strong and assertive. No valid conclusions based on the results are made. Please revise. 

Answer) I corrected as follows

In conclusion, the results of this investigation indicated a correlation between frailty and asthma in the elderly. Frailty may not only be asthma but also may other diseases too. So more evidences were needed to establish this association. But this finding might help establish policies how reduce the bur-den of asthma and frailty in the elderly.

Minor points

  1. Please, comment on the limitations of the study - how can they impact on your outcomes

Answer) I added limitation as follows

Finally K-FRAIL scale is used to assess frailty risk rather than to diagnose frailty, so it is not a diagnostic tool for frailty. Therefore, caution is necessary when interpreting our results regarding the association between elderly asthma and frailty. Further well-designed studies are needed to elucidate the relationship between frailty and elderly asthma.

  1. Could you please discuss the clinical implications of the results and what recommendations could be made based on the results?

Answer) I added as follows

But this finding might help establish policies how reduce the burden of asthma and frailty in the elderly.

  1.         The abstract should be one paragraph (please revise), structured (this is done)

 Answer) I corrected

4 The conclusion in the abstract should be revised since there is no such term as "Korean asthma" or another nationality-asthma. I suggest something like "in asthma patients from the Korean population", etc.

 Answer) I corrected

  1. Lne 44-47 - COPD has been mentioned out of the blue; please focus on asthma OR, if you include COPD, explain in detail why you are meeting it.

 Answer) I corrected

  1. The hypothesis in the introduction should be extended. Also, the aim should be more precise and detailed. 

 Answer) I corrected

.

  1. Author contribution is not filled. 

Answer) I corrected

  1. State the Institutional Review Board Statement at the end of the manuscript

Answer) I corrected as follows

Reviewer 3 Report

Comments and Suggestions for Authors

Dear Author/s
It has been a pleasure to read your current paper.
The manuscript is well structured and includes usefull information on the assesment of frailty
in elderly patients with asthma.

Your study is well documented and addresses a useful topic in pulmonology and also in general practice, namely the functional physiological reserve and body performance of patients with a chronic condition during aging.

The study design is described in details, the methods are also presented quite clear. I appreciated that the authors included a control group.

In the introduction in lines 46-47 there is a mention on a prior study on the association of frailty in asthma patients. This study should be mentioned and added in the reference list.

Lines 40-43: I appreciated that the authors mentioned the standardized questionnaires that was used, and that this Frailty scale was previously translated and validated in the Korean population, before they were applied it to patients. I would suggest to refer this aspects in the „Methods” section, instead in the „Introduction”.

In the methods section in lines 63-69 it is not clear if the authors underwent any prospective study or if they collected data from the Korea Institute for Health and Social Affairs and the Korean Ministry of Health and Welfare, that were already available.

Lines 69-71: this was the only criteria for inclusion in the study group?

Lines 82-83: is not clear what the authors want to say with this statement. Please clarify. The non-asthma group included patients with COPD? This could be a major bias.

Lines 84-85: „We divided BMI into four groups” – probably the study group (or both asthma group and control group) were divided into four groups according to BMI

Lines 92-95: the MMSE-KC was applied prospectively by the authors in each candidates from the both groups (asthma and control)?

Line 98: who created the geriatric depression scale (GDS)?

Lines 105-107: the GDS scale was applied prospectively by the authors in each candidates from the both groups (asthma and control), so they could be assigned to the new groups mentioned?

Lines 146-147: For data analysis the software that was used is not mentioned.

In the results, the socio-demographic and clinical characteristics between the asthma and non-asthma group are very well synthesized. There are a few aspects that should be clarified in this table:

1. It is not clear what „religion yes/no” means

2. in „Chronic illness” for the asthma group – does this mean other comorbidities beside asthma?

Table 3 is not a real table. The results should only be presented in lines 161-162 if there is no other data that this table could include.

The discussion section provides valuable comments on the combined mechanisms of frailty in asthma patients. An important aspect that was not addressed in this study is the rate of exacerbations and hospitalizations, as mentioned in lines 186-190. This are aspects that should be added to the limitations of this study.

I appreciated the concise structure of the paper, the comparative data within the two groups that were analysed. All the relevant aspects regarding the characteristics of asthma patients are sumarized very well in table 1. Table 2 is very concise and clear.

, it may contribute to a better awareness on the  burden of asthma including frailty in elderly patients.

With all best wishes

Comments on the Quality of English Language

Some of the statements are difficult to understand and the format of some phrases is wrong. For example lines 84-85.

Author Response

Dear Author/s
It has been a pleasure to read your current paper.
The manuscript is well structured and includes usefull information on the assesment of frailty in elderly patients with asthma.

Your study is well documented and addresses a useful topic in pulmonology and also in general practice, namely the functional physiological reserve and body performance of patients with a chronic condition during aging.

The study design is described in details, the methods are also presented quite clear. I appreciated that the authors included a control group.

In the introduction in lines 46-47 there is a mention on a prior study on the association of frailty in asthma patients. This study should be mentioned and added in the reference list.

Lines 40-43: I appreciated that the authors mentioned the standardized questionnaires that was used, and that this Frailty scale was previously translated and validated in the Korean population, before they were applied it to patients. I would suggest to refer this aspects in the „Methods” section, instead in the „Introduction”.

Answer) I corrected

In the methods section in lines 63-69 it is not clear if the authors underwent any prospective study or if they collected data from the Korea Institute for Health and Social Affairs and the Korean Ministry of Health and Welfare, that were already available.

Lines 69-71: this was the only criteria for inclusion in the study group?

Answer) Yes

Lines 82-83: is not clear what the authors want to say with this statement. Please clarify. The non-asthma group included patients with COPD? This could be a major bias.

Answer) We does not included COPD We added more clearly.

Lines 84-85: „We divided BMI into four groups” – probably the study group (or both asthma group and control group) were divided into four groups according to BMI

Answer) We analyzed BMI as confounding variables

Lines 92-95: the MMSE-KC was applied prospectively by the authors in each candidates from the both groups (asthma and control)?

Answer) Our study investigated MMSE-KC cross-sectionally.  

Line 98: who created the geriatric depression scale (GDS)?

Answer) I added reference

Lines 105-107: the GDS scale was applied prospectively by the authors in each candidates from the both groups (asthma and control), so they could be assigned to the new groups mentioned?

Answer) Our study investigated GDS cross-sectionally.  

Lines 146-147: For data analysis the software that was used is not mentioned.

Answer) I corrected

In the results, the socio-demographic and clinical characteristics between the asthma and non-asthma group are very well synthesized. There are a few aspects that should be clarified in this table:

  1. It is not clear what „religion yes/no” means

Answer) I corrected

  1. in „Chronic illness” for the asthma group – does this mean other comorbidities beside asthma?

Answer) I corrected

Table 3 is not a real table. The results should only be presented in lines 161-162 if there is no other data that this table could include.

Answer) I corrected

The discussion section provides valuable comments on the combined mechanisms of frailty in asthma patients. An important aspect that was not addressed in this study is the rate of exacerbations and hospitalizations, as mentioned in lines 186-190. This are aspects that should be added to the limitations of this study.

Answer) I corrected as follows

Besides, we did not assess current asthma symptoms and corticosteroid usage, rate of asthma exacerbations and hospitalizations due to asthma.

I appreciated the concise structure of the paper, the comparative data within the two groups that were analysed. All the relevant aspects regarding the characteristics of asthma patients are sumarized very well in table 1. Table 2 is very concise and clear.

, it may contribute to a better awareness on the  burden of asthma including frailty in elderly patients.

Answer) I corrected

With all best wishes

Round 2

Reviewer 2 Report

Comments and Suggestions for Authors

Dear Authors,
I have been carefully reviewed your revised article. In my opinion, this revised manuscript is fully acceptable.

Comments on the Quality of English Language

Fine